## PERSPECTIVE

### Plasticity reloaded: motor skill learning plus stimulation boosts adaptations in the ageing nervous system

**Wolfgang Taube** (iD)

*Department of Neuroscience and Movement Science, University of Fribourg, Fribourg, Switzerland*

Email: wolfgang.taube@unifr.ch

Handling Editors: Richard Carson & Uros Marusic

The peer review history is available in the Supporting Information section of this article (https://doi.org/10.1113/JP288884#support-information-section).

In 1928 Nobel Prize winner Santiago Ramón y Cajal stated that once brain development is complete, the mechanisms that drive the growth and regeneration of axons and dendrites are permanently lost (Cajal 1928). According to this view the adult brain's neural pathways were fixed and unchangeable – capable of dying, but not of regenerating. For a long time it was believed that the adult CNS lacked plasticity altogether, with no possibility for adaptation or structural change beyond early development.

Later research, however, revealed that young adults still exhibit neuroplasticity and even neurogenesis, especially when learning new motor skills. Still it was commonly assumed that in older adults, such plasticity was extremely limited. At best it was believed that engaging in demanding physical activities might only slow the inevitable decline associated with ageing.

Today we understand that the CNS retains its capacity for change even in older age, though the process may be slower compared to younger individuals. The question now is not whether the elderly brain can adapt but rather what promotes these adaptive changes? A healthy diet, regular physical activity, good quality sleep, a balanced lifestyle and strong social connections all contribute positively. However some artificial help using new technological interventions may be beneficial too – and this is the focus of the following discussion.

A research group led by Jesper Lundbye-Jensen explored how well older adults can adapt when practicing a ballistic motor task (Bjørndal et al., 2026). It is well established that the ability to produce force rapidly declines more quickly with age than the capacity to produce maximal force. Yet this rapid force generation is crucial for reacting to stumbles, trips or slips – and thus, for preventing falls. Therefore it is essential to explore training strategies that can help older adults improve ballistic force production. In their study, elderly adults displayed lower maximal acceleration compared to young adults and, more importantly, smaller improvements with practice, indicating inferior learning and lower ability to retain the effects obtained through motor practice (i.e. consolidation) compared to young healthy adults. However motor learning and motor memory consolidation could be improved by repeatedly activating the synaptic transmission between cortical and spinal motoneurons by applying a 'paired corticospinal-motoneuronal stimulation' (PCMS) protocol prior to motor practice. This is consistent with a recent proof-of-principle study in young adults (Bjørndal et al., 2024). In this protocol descending volleys elicited by transcranial magnetic stimulation (TMS) were timed to arrive at the cortico-motoneuronal presynapse 2 ms before the arrival of antidromic volleys at the postsynapse evoked from peripheral nerve stimulation. The PCMS procedure was repeated 100 times before the participants had practiced the ballistic task. Interestingly PCMS did not only improve performance in the immediate retention test performed on the same day but also 7 days later; i.e. it improved both short- and long-term consolidation. At the same time neurophysiological measures were applied to investigate the effect of PCMS on neural markers. PCMS increased corticospinal excitability, probably by altering connectivity at the corticomotoneuronal synapse, i.e. at the spinal level.

The fundamental mechanism by which neural connectivity can be altered at the cellular level has been well known for over 75 years: 'WHAT FIRES TOGETHER, WIRES TOGETHER' (Hebb, 1949). Depending on the timing of pre- and post-synaptic activation, synaptic transmission can be either facilitated or inhibited. This phenomenon, known as use-dependent plasticity, can be triggered through motor skill learning, or by artificially stimulating pre- and postsynaptic neurons – and, as this study elegantly demonstrates, by combining both.

One intriguing aspect that warrants further exploration is whether 150 ballistic repetitions combined with 100 PCMS pulses yields the same results as 250 ballistic repetitions, or if the processes underlying motor practice and paired stimulation are subtly different and potentially complementary. Additionally future studies could focus on long-term changes in synaptic transmission at the spinal level. Although motor learning processes in the cortex are relatively well understood with early (e.g. long-term potentiation) and late(r) phases (e.g. synaptogenesis) demonstrating distinct neural plasticity processes, adaptations at the spinal level, particularly accompanying PCMS, remain less explored. This area holds considerable promise for therapeutic applications, especially in the rehabilitation of spinal cord injuries (Christiansen & Perez, 2018).

In summary the study by Bjørndal and colleagues offers compelling behavioural evidence that PCMS can enhance ballistic motor learning and motor memory consolidation in older adults. Neuro-physiological measures of neural plasticity also suggest that PCMS produces superior outcomes. The paper thus stands out not only for its robust basic science contribution but also for its significant potential for real-world application, particularly in promoting neural plasticity in both ageing populations and patients. It is hoped that this work will inspire further advancements in the field, helping to unlock new therapeutic strategies for rehabilitation.

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

## Additional information

### Competing interests

There are no competing interests.

### Author contributions

W.T.: Conception or design of the work; drafting the work or revising it critically for important intellectual content; final approval of the version to be published; agreement to be accountable for all aspects of the work.

### Funding

This work was not supported by any funds.

### Acknowledgements

Open access publishing facilitated by Universite de Fribourg, as part of the Wiley - Universite de Fribourg agreement via the Consortium Of Swiss Academic Libraries.

### Keywords

elderly, Hebbian priming, motor skill learning, paired corticospinal-motoneuronal stimulation

## Supporting information

Additional supporting information can be found online in the Supporting Information section at the end of the HTML view of the article. Supporting information files available:

**Peer Review History**

