## [Peer Review History · The Journal of Physiology]

Plasticity Reloaded: Motor skill learning plus stimulation boosts adaptations in the aging nervous system

Wolfgang Taube

DOI: 10.1113/JP288884

Corresponding author(s): Wolfgang Taube (wolfgang.taube@unifr.ch)

Review Timeline:

Submission Date:

15-Apr-2025

Accepted:

30-Apr-2025

Senior Editor: Richard Carson

Reviewing Editor: Uros Marusic

Transaction Report:

Dear Professor Taube,

Re: JP-P-2025-288884 "**Plasticity Reloaded: Motor skill learning plus stimulation boosts adaptations in the aging nervous system**" by Wolfgang Taube

We are pleased to tell you that your paper has been accepted for publication in The Journal of Physiology.

Yours sincerely,

Richard Carson
Senior Editor
The Journal of Physiology

If you would like to receive our 'Research Roundup', a monthly newsletter highlighting the cutting-edge research published in The Physiological Society's family of journals (The Journal of Physiology, Experimental Physiology, Physiological Reports, The Journal of Nutritional Physiology, and The Journal of Precision Medicine: Health and Disease), please click this link, fill in your name and email address and select 'Research Roundup':

<https://www.physoc.org/journals-and-media/membernews>

- You can help your research get the attention it deserves! Check out Wiley's free Promotion Guide for best-practice recommendations for promoting your work at: www.wileyauthors.com/eeo/guide. You can learn more about Wiley Editing Services which offers professional video, design, and writing services to create shareable video abstracts, infographics, conference posters, lay summaries, and research news stories for your research at: www.wileyauthors.com/eeo/promotion.

The Corresponding Author will receive an email from Wiley with details on how to register or log-in to Wiley Authors Services where you will be able to place an order

EDITOR COMMENTS

Reviewing Editor:

We are pleased to inform you that your Perspective paper, "Plasticity Reloaded: Motor skill learning plus stimulation boosts adaptations in the aging nervous system," has been accepted for publication in The Journal of Physiology. The reviewers found your novel viewpoint on enhancing neuroplasticity in aging through combined motor learning and stimulation to be insightful commentary and thus a valuable contribution to the field.

REFEREE COMMENTS

Referee #1:

The perspective paper by Wolfgang Taube is well-written and factually accurate. I have no further comments or corrections.